# Temporal Visual Patterns of Construction Hazard Recognition Strategies

**DOI:** 10.3390/ijerph18168779

**Published:** 2021-08-20

**Authors:** Rui Cheng, Jiaming Wang, Pin-Chao Liao

**Affiliations:** 1Department of Construction Management, Tsinghua University, Beijing 100084, China; chengr19@mails.tsinghua.edu.cn; 2School of Economics and Management, Tongji University, Shanghai 200092, China; 1551288@tongji.edu.cn

**Keywords:** construction hazard recognition (CHR), temporal qualitative comparative analysis (TQCA), visual patterns, computer vision, construction safety

## Abstract

Visual cognitive strategies in construction hazard recognition (CHR) signifies prominent value for the development of CHR computer vision techniques and safety training. Nonetheless, most studies are based on either sparse fixations or cross-sectional (accumulative) statistics, which lack consideration of temporality and yielding limited visual pattern information. This research aims to investigate the temporal visual search patterns for CHR and the cognitive strategies they imply. An experimental study was designed to simulate CHR and document participants’ visual behavior. Temporal qualitative comparative analysis (TQCA) was applied to analyze the CHR visual sequences. The results were triangulated based on post-event interviews and show that: (1) In the potential electrical contact hazards, the intersection of the energy-releasing source and wire that reflected their interaction is the cognitively driven visual area that participants tend to prioritize; (2) in the PPE-related hazards, two different visual strategies, i.e., “scene-related” and “norm-guided”, can usually be generalized according to the participants’ visual cognitive logic, corresponding to the bottom-up (experience oriented) and top-down (safety knowledge oriented) cognitive models. This paper extended recognition-by-components (RBC) model and gestalt model as well as providing feasible practical guide for safety trainings and theoretical foundations of computer vision techniques for CHR.

## 1. Introduction

Hazard recognition is the first step to prevent accidents [1]. In recent years, computer vision-based construction hazard recognition (CHR) technology has attracted widespread interest of researchers [2], such as identifying and tracking physical elements (including temporary structures, machine and materials, etc.) [3,4,5], and capturing unsafe behaviors of workers [6,7]. In fact, the basic logic of computer vision technology algorithms for the purpose of identifying hazards is mostly based on humans’ psychological process of hazard cognition [8].

However, there is still a distance to achieve applying computer vision technology instead of manual inspection for CHR in construction safety management. The hazard recognition performed by humans is a high-level cognitive process [8,9]. Construction workers detect sensory signals from the environment, selectively pay attention to them, and match the perceived information with safety knowledge to make judgments about the presence of hazards [10]. This means that construction workers can determine potential hazards that may occur in the future based on their observation and understanding of the current scene. However, this is a challenge for computer vision because it requires computers to have similar cognitive abilities as humans, i.e., to understand the meaning of scenes and generate new information by association to drive reasoning and recognition of unknown sets of potential hazards [2]. As a consequence, in order to further promote the development and application of computer vision technology in construction safety management, it is necessary to fully understand humans’ cognitive processes and strategies of CHR.

The nature of the visual search process is an important and typical physiological activity for hazard recognition [11]. Previous studies have tried to characterize the visual patterns during CHR through a series of eye-movement indicators [1]. Among them, the most commonly used are area-of-interest (AOI)-based indicators such as fixation times [12], duration of fixation [13] and fixation space density [14]. Although these studies have demonstrated the existence of patterned commonalities in people’s visual behaviors and extended the understanding of visual patterns, relying only on descriptive statistics of common features is still insufficient to describe the specifics of visual patterns. Because visual patterns represent dynamic visual processes that last for a certain amount of time, rather than transient states. In recent years, researchers have attempted to describe visual patterns by recording changes in the spatial location characteristics of visual fixations and thus generating visual scan paths [15,16], which holds promise for studying the human cognitive processes and strategies implied by visual patterns. However, due to the deficiencies in the analytical methods employed in these studies in terms of the ability to process visual sequence data, they focused only on cross-sectional data and selectively or forcibly ignored the temporal nature of visual patterns, or have stuck to the stage of making overall comparisons of visual sequences, resulting in a lack of sufficient details to identify and interpret the visual patterns implied in these sequences [16,17]. This also presents a significant gap for researchers to further exploring the cognitive strategy that motivate people to choose to perform certain visual patterns. Therefore, more in-depth studies and appropriate visual sequence analysis methods are needed to help decipher visual patterns and thus further explore visual cognitive strategies of CHR.

In summary, the core research questions of this paper are: (1) What are the visual patterns employed by those who successfully identify hazards during their visual search in CHR? (2) Why are such visual patterns employed and what cognitive strategies do they imply?

The research objectives are (1) to design and conduct an eye-tracking experiment with real construction scenes pictures as the CHR experimental background to document the participants’ visual sequences during their CHR processes; (2) to introduce TQCA method with the analytical capability for temporality which allows for the inclusion of sequence conditions, and apply it in the visual sequences analysis to identify the visual patterns for CHR; (3) to interpret the visual patterns from both quantitative and qualitative perspectives and explore the visual cognitive strategies of CHR implied by these visual patterns. Further, the practical contributions of the findings to computer vision and safety training for CHR as well as the theoretical contributions to visual cognitive psychology theories are discussed.

The rest of the paper is structured as follows: Section 2 illustrates the existing research gaps for the study. Section 3 lays out the methodology and the experimental design. Section 4 presents the results and the data analyses. Section 5 discusses how the results complement the research gaps and clarify the contributions of this study. Finally, the last section summarizes the main findings of this study, notes its limitations and suggests the scope for future research.

## 2. Literature Review

### 2.1. The Development of Computer Vision in CHR: Grounded in Human Cognitive Mechanisms

In the last decade, the application of computer vision in construction safety management has become a hot frontier of research [2], with algorithms mostly grounded in cognitive psychology models [8]. The template matching model (TM) claims that individual things can be compared with established criteria to obtain results; the recognition-by-components model (RBC) argues that the whole can be described and identified based on the relationship and interaction between its local components [18]. A good example is the discrimination of human behavior and posture in construction sites. First, the presence of a person can be determined based on elements such as head, limbs, etc. (RBC). Then, the presence of hidden hazards can be judged by comparing human posture and wear with safety norms (RBC + TM), for example, identifying unsafe behaviors and abnormal postures based on skeletal movements [19,20] and determining whether workers are wearing personal protective equipment (PPE) such as helmets and safety belts [7,21,22]. Even though such algorithms are computationally efficient and reliable, most of them discriminate low visual complexity hazards with a fixed perspective, which is difficult to adapt to complex and dynamic construction scenarios.

Some studies claim that hazards can be identified by specifying some distinguishing features, which is also connoted by feature matching models (FM) [18]. Such studies extract and compare scene components (people [23], materials [24], and machinery [25]) with the template or features described by the norm to determine the presence of hazards [26]. For instance, research has been implemented to identify machinery and materials in construction sites by features (e.g., histograms of oriented gradients and colors (HOG + C)) [23], calculate distances and compare them to thresholds to identify the hazards. Another example is the determination of hazards based on the dynamic interaction of objects under spatial-temporal motion, such as the movement of excavators [27] and the spatial proximity to people (Kim, Liu et al., 2019). Although such studies take into account the temporal and dynamic nature and broaden the approaches of safety management, the accuracy of hazard recognition remains a major concern. The main reasons for this include: (1) the lack of basic data for training algorithmic models; and (2) the reliance on inefficient and expensive manual annotation (components of hazards, etc.) for the generation of available data [28]. In addition, selecting the appropriate content in many safety regulations with their inherent complexity and then accurately mapping it to the appropriate specific components of the construction scenario is quite time-consuming and difficult [29,30].

It has been found that human attention is cognitively driven to differentially allocate between visual areas and shift over time. The lack of such a mechanism means that computers cannot learn and utilize human attentional cues, which is one of the important factors limiting the development of computer vision technology for CHR [31]. On the other hand, computer vision for CHR that lays the foundation of the above cognitive models usually judges only based on the existing observable scene components, and cannot combine the association and inference of possible future states to further infer whether the existing scene has the possibility of hazard occurrence. Therefore, how can interactions between components and transitional states between scenes be identified? How to determine the possible hazards of the current state in the future based on associations? Understanding how humans solve these problems in the cognitive process would provide important implications for the further development of computer vision algorithms for CHR.

The completeness model suggests that humans can recognize objects with the same components and structure based on the relationship of the components and the degree to which the relationship “deviates from the prototype” or “average” [18]. In other words, computer vision techniques for hazard recognition need to integrate concepts such as components, local/global features and “prototypes” to be fully effective. Therefore, research in the construction industry needs to further complement the mechanism of human cognitive processes for hazards based on these models.

### 2.2. Methodological Deficiencies and Limitations in CHR Visual Patterns Studies

Many studies of construction industry have reached a consensus to consider hazard recognition as a visual search task [14]. By documenting visual behaviors during hazard recognition [32], it is possible to study construction workers’ visual strategies for CHR and improve the utilization ability of attentional cues [31], thus further complementing the logical basis for the development of computer vision techniques and the improvement for safety training. However, synthesizing current research, this research direction still faces three main dilemmas.

Firstly, most previous studies have generalized the characteristics and commonalities of visual patterns through descriptive statistics of AOI-based eye movement indicators, such as fixation count [12,33], fixation duration [14,34], and the heat map [35]. For example, studies have attempted to use fixation-related indicators to measure participants’ attention allocation [36] and situational awareness [37]. However, these studies were insufficient to describe the specific content of visual patterns and ignored the temporal nature of visual cognitive processes, making them fail to accurately answer the fundamental question of what visual patterns people employed in performing visual cognitive tasks. On the one hand, visual patterns represent a dynamic visual process that lasts for a certain amount of time, rather than a transient state or a static segment. Therefore, relying on holistic indicators that lacks the key attribute of temporality cannot adequately express or define the substance of the visual pattern. On the other hand, quantitative analysis based on the full amount of raw visual data faces problems such as excessive noise, limited analytical power and applicability, so previous studies have mostly used qualitative methods such as descriptive summary and comparison to slice, sample or analyze visual processes as a whole [13]. This has also led to the inability of existing studies to capture the temporal nature of visual scanning paths, whose results were limited by statistical analysis of cross-sectional data [16] and indicators.

Secondly, the robustness of these AOI-based eye-movement indicators is vulnerable to the impact of the accuracy of AOI definitions. Many studies only labelled AOIs according to expert opinions or the subjective opinions of researchers, which implies a further decrease in these indicators’ reliability and validity of the studies [16]. In order to overcome the subjectivity problem of traditional AOI pre-definition methods, some studies have also attempted to adopt data-driven AOI definition methods, such as fixation clustering [16,38,39]. However, the AOI based on data-driven determination may be affected by factors such as subjects’ cryptic recognition ability and time-varying attention, leading to potential unpredictable accumulation of errors, thus making the blurring of AOI boundaries and the robustness of the associated indicators questionable.

Finally, although analyzing visual search patterns based on temporal and spatial sequences in visual scanning paths has been demonstrated to be a valid and appropriate approach and has potential in helping to understand the cognitive process of visual search [16,17], current research on visual sequences in visual scanning paths is still relatively superficial. The essence of visual scanning paths sequence analysis is to analyze the visual transition sequences between AOIs within a certain time range in a longitudinal manner. By using characters to represent the AOI where the fixations are located, the expression of visual scanning path can be simplified as strings, which allows for quantitative analysis based on fixed visual sequences [16,40]. For example, Xu, Chong et al. performed an overall similarity analysis between the visual sequences of participants who successfully identified hazards with those who failed and found that successful participants followed a similar visual pattern of hazards searching, i.e., had similar visual sequences [16]. Although this implied that the mental representations and cognitive strategies were of significant research interest, the research did not involve the specific content of the sequences. Studies limited to overall cross-sectional comparisons of sequences cannot help address specific hazard scenarios due to the lack of ability to interpret why people adopt specific visual cognitive strategies, thus failing to provide substantial assistance in the design of safety training and the development of hazard recognition aids such as computer vision-based safety management techniques.

In fact, one of the main reasons for the above dilemma lies in the lack of appropriate analytical methods that can be applied to visual sequence analysis incorporating temporality. Current research has gradually recognized the importance of temporality for the study of visual patterns and attempted to address this issue. For example, Chong, Liang [17] used neural networks to pre-analyze the basic visual searching sequences of two hazards, namely “potential electrical contact” and “struck-by hazards”, obtained several visual sequence fragments (the character strings formed by fixation transfer sequence between AOIs) with different weights, and finally used it as basic conditions for the subsequent crisp-set QCA configuration analysis to summarize the visual search patterns in the corresponding hazard scene. Despite the advancement of visual path research, there are still two unavoidable problems: (1) The sequences contain different lengths of time, which reduces the possibility of identifying reliable CHR visual patterns; (2) QCA requires the selection conditions and cases with equal weights, but the calculation of neural networks gives different weights to visual sequence segments (conditions), which violates QCA’s basic requirements for conditional equality, leading to the results with individual bias. Therefore, it is necessary to adopt more advanced analysis approaches to extend the limited analysis and interpretation capabilities of visual sequences. 

## 3. Methodology

### 3.1. Technical Route

As shown in the study framework in Figure 1, this study was conducted in two parts. First, an experiment of a hazard recognition task was designed and tested using an advanced eye-tracking device to record participants’ visual behavior data. Subsequently, a TQCA method was used to analyze the data from the visual behavior sequences in an attempt to generalize the visual strategies for hazard recognition.

### 3.2. Experimental Design

#### 3.2.1. Experimental Protocol

We conducted a hazard recognition experiment in which participants were asked to determine whether the observed pictures of construction scenes were hazardous or safe. All photo material used in the study was collected from construction sites. A total of 60 sets of construction site pictures (60 for hazardous conditions and 60 for safe conditions, 120 in total) were obtained from a pre-developed database that compiled pictures of scenes showing hazardous situations and after rectification (safety). The formal experiment was divided into two parts. Part A: observe 120 pictures that appeared in random order in sequence and complete the judgment according to the paradigm shown in Figure 2; Part B: observe and respond again to 3 pre-selected dangerous and 3 safe pictures, all 6 pictures were selected from the 120 pictures in Part A. A 5-min break was set between Part A and Part B to relieve participants’ possible eyestrain. Furthermore, and then, the answers of Part A and Part B were compared for consistency to aid in identifying unreliable participants.

A total of 85 Han Chinese male construction workers with an average age of 42.2 were recruited from construction sites in Beijing, whose ages ranged from 21 to 60 years. All participants participated in industry-specific safety training and were educated in hazard recognition, and the visual process of 69 workers was completely recorded. Further, 14 workers’ data were ultimately excluded to ensure data quality, as their responses were considered unreliable. When judging the scenes shown in the pictures as dangerous or safe, these 14 individuals chose the same response result for more than 90% of the pictures, which could be due to the non-serious response attitude or limited comprehension of the experimental rules. Therefore, the researchers finally selected a sample of 55 male participants for visual behavior analysis. All participants had normal or corrected normal vision with similar work experience in construction projects. This study was approved by the local ethics committee of Tsinghua University (No. 201914). All participants provided written informed consent and received monetary compensation.

#### 3.2.2. Explanation of the Experimental Protocol

This proposal is closer to the real conditions of the construction scenario. Since construction workers generally live and work on construction sites for a long time, they have prior observation and comprehension of the construction site without having to be familiar with the environment when performing hazard recognition. However, the 120 pictures selected for this study were from different construction projects in which the participants had never participated. If the participants were directly asked to recognize hazards in the unfamiliar environment, they would have to spend some time familiarizing themselves with the construction scene first after seeing each picture, which might reduce the effectiveness of the visual search sequence obtained from the experiment. As a result, we designed the experiment by utilizing the psychological principles of “working memory” and “priming effect” to give the participants a certain initial impression of the construction site by quickly showing 120 pictures, avoiding the bias caused by spending a long time observing the environment. After that, we selected three pictures with hazards from the 120 pictures for Part B, and let the participants answer under the condition that they were initially familiar with the construction scene. It is worth emphasizing that after the completion of Part A, the participants were not given any information about the correctness of their responses, so the reliability of the responses and experimental data in Part B were not affected by Part A. 

In addition, this study relied on a large systematic experiment, and the data sources needed in this study were focused on Part B, that is, Part A was not significant for this study. Therefore, for this experiment, the real purpose of Part A was to help the participants become familiar with the construction scenario rather than to use the answers and data from Part A for subsequent analysis or for comparison. In this experiment, the purpose of asking the participants in Part A to answer was to make them concentrate on the construction scenes and memorize them, so as to avoid any possible negligence after being directly told to “just browse”. In summary, the above design can help us to draw conclusions closer to the real construction scene.

In Part B, because of the vague impression of the scenes left by the first observation, when participants were asked to observe again, they claimed to perform the hazard recognition task in a more detailed and strategic manner. Examination of the video recordings of visual behavior confirmed the reliability of this statement. In 1974, based on experiments simulating short-term memory impairment, Baddeley and Hitch came up with the concept of working memory as the ability to store information for short periods of time and use the information for cognitive activities such as processing, manipulation, or reasoning when cognitively required [39]. In fact, working memory also belongs to short-term memory, but it emphasizes the association of short-term memory with the job in which the person is currently engaged. In this experiment, the content of short-term memory kept changing and showed some systematicity due to the need of the hazard recognition task. On this basis, short-term memory formed a continuous system over time, i.e., working memory. The visual cognitive strategy was a manifestation of this system, which effectively helped the participants to complete the visual search task [41].

On the basis of “working memory” and “short-term memory”, we can further derive the “priming effect” in psychology. The priming effect refers to a psychological phenomenon in which the perception and processing of a stimulus becomes easier because of the previous influence of the stimulus. Some researchers believe that the priming effect is a manifestation of implicit memory. Accurately speaking, a direct priming effect was involved in this experiment because the stimuli presented before and after were identical. It can be inferred that the working memory formed in Part A had a reinforcing effect on the visual behavior and strategy in Part B, which achieved the purpose of helping the participants to be familiar with the construction scene previously and reduce the bias of the visual search sequence.

### 3.3. Post-Experiment Verification

After the experiment was completed, the researchers interviewed the participants. All participants were asked to review the scenes they judged as hazardous in Part B and describe their observation process as well as the basis of their judgment in detail. For consistency validation, the experimenters recorded and organized the participants’ responses, and then carefully studied the participants’ observation process and visual hotspot diagrams. A total of 55 samples showed high consistency between visual behavior and personal representations.

### 3.4. Data Analysis

#### 3.4.1. Sequence Generation

Eye movement data were collected using Tobii Pro Fusion, which recorded at a sampling frequency of 120 Hz. Prior to recording, Tobii Pro Fusion was connected to Tobii Pro Lab software, an integrated software used to design and perform the experiment process and analyze the eye movement data, thus performing calibration to ensure that the camera on the eye-tracking device (Tobii Pro Fusion) accurately tracked the participant’s eyes. The Tobii Pro Lab software has a live viewing function that allows scenes recorded by the Tobii Pro Fusion to be viewed and tagged in real time. Thus, the full course of the participant’s visual trajectory and fixations will be recorded in time and easily found during replay.

The raw eye-movement data processing is performed in the Tobii Pro Lab, which allows direct viewing and analysis of the visual behavior data recorded by Tobii Pro Fusion. Finally, for each hazard, the location data of all mapped fixations are output sequentially, and their coordinates are expressed in pixels. The researchers define the locations and ranges of the 3 AOIs based on the descriptions of each hazard in the database, and name them A, B and C. Based on the position of its coordinates in relation to the AOI, it is possible to determine whether the gaze point is located inside the AOI. Based on the coordinates of a fixation in relation to the inclusion of the AOI area in the picture, it can be determined whether it is located inside the AOI. Visual scan paths are generated and represented as a fixed sequence of AOIs. Specifically, the scan path is a string that sequentially shows the AOIs that the observer gazes at in chronological order, i.e., each fixation that makes up the visual sequence is represented by the letter corresponding to the AOI to which it belongs. 

West et al. suggested two forms of fixation sequences to suit different research focuses [42]. The extended sequence represents each gaze point as a letter to form a string, while the collapsed sequence replaces consecutive strings composed of the same letters with a single letter to highlight transitions between AOIs (e.g., the extended sequence “AABB” can be collapsed to “AB”). Following West’s suggestion, this study adopted collapsed sequences in an attempt to generalize the visual pattern, i.e., using the first fixation that falls within an AOI to generalize all subsequent fixations within the same AOI, which implies that de-duplication will be required in the subsequent data preparation.

#### 3.4.2. TQCA Method

In recent years, there has been an increasing interest in the study of configuration, namely “multidimensional constellations of conceptually distinct characteristics that occur together” [43]. Typically, such studies focus on a process, i.e., a series of activities that unfold over time [44]. Qualitative comparative analysis (QCA) is considered a good solution for analyzing such problems and has been spread and applied in several disciplines, describing how a system can reach a certain outcome from different initial conditions through different or multiple paths [45]. Therefore, QCA method is widely used with rapid development. However, QCA has often been criticized for its static nature and limitations, as time course and change are essential to understanding the complex phenomena of a case [46]. Traditional QCA assumes that the conditions are equivalent, and variables as well as cases are “frozen in time”. Researchers try to combine the QCA method with timeliness, and De Meur, Rihoux [47] listed five solutions to deal with time sequence in QCA, of which the relatively mature method is the temporal qualitative comparative analysis (TQCA) proposed by Caren and Panofsky [48]. To sum up, TQCA is an extension of QCA that is applicable to study the causal conditions occurring in the sequence [49].

In this study, we believe that TQCA is the appropriate method for this study, because (1) TQCA extends the analytical capability for temporality while retaining the advantages of the traditional QCA method, which makes TQCA advantageous in applying to model generalization and integrated synthesis of complex processes involving multiple conditions and time sequences (e.g., the CHR process in this study) from both quantitative and qualitative perspectives; (2) based on the TQCA analysis of the participants’ visual searching paths, the visual search configurations with high commonality could be identified, thus helping to address the research question of “What are the visual patterns of construction workers during CHR?”; (3) TQCA allows for the inclusion of sequence conditions in the analysis and preserves the integrity of the sequence conditions in both the analysis process and output results, which helps the researcher to interpret the content of the visual patterns in order to further explore and generalize cognitive strategies.

In order to conduct sequences analysis in the TQCA method by existing QCA analysis software and algorithms, this study encoded the observed sequence of any two AOIs as a sequence condition variable [44] and generated 6 temporal sequence conditions. Specifically, if the participant observed A first and then B, then condition “AB” was coded as “1” and “BA” was coded as 0, and vice versa. However, it should be noted that if there was no observed AOI in A and B, which means no sequence exists between A and B, so “AB” and “BA” were both coded as 0. In this way, all observation sequences of any participant were uniquely determined without the possibility of duplication or omission. In addition, the new sequence truth table generated was able to be analyzed by existing QCA analysis tools, such as fsQCA3.0.

This study, mainly composed of two parts including necessity and sufficiency analysis that formed the basis of configuration path analysis, used fsQCA3.0 software for TQCA analysis. Necessity means that when the result is produced, a condition always occurs. Sufficiency refers to the explanatory power of a condition in explaining the occurrence of a result [50]. Consistency indicates the ability of a condition to lead to an outcome. In the necessity analysis phase (conditional necessity test), a condition can be considered necessary for an outcome when it satisfies a minimum threshold of consistency (≥0.9) [51]. In the sufficiency analysis phase (configuration analysis), the quality of the visual pattern can be evaluated in terms of PRI (proportional reduction of inconsistency, explaining the consistency of subset relations) between 0 and 1, based on the output provided by the QCA method [52]. In general, a grouping with a PRI value above 0.75 represents an acceptable and effective pattern [51].

For the final result, QCA can produce three types of solutions: complex, parsimonious, and intermediate. This study uses intermediate solutions for a complete representation of the visual pattern in the hazard recognition process. It should be noted that in order to avoid possible interference of the results by chance observation cases caused by individual participant bias and considering the size of the sample size (55 cases), this study set the minimum acceptable frequency cutoff threshold set at 2 cases per configuration and the minimum acceptable PRI consistency cutoff threshold at 0.75. Configurations that do not meet these requirements will be neglected.

## 4. Results

### 4.1. Descriptive Statistics

As shown in Table 1, 3 pictures selected from the 120 pictures correspond to 3 hazards respectively, and the 3 hazards belong to 2 hazard types. Hazard 1 (Figure 3), whose visual heat map is shown in Figure 4, is associated with potential electrical contact, while Hazard 2 (Figure 5) and Hazard 3 (Figure 6) are associated with the failure of personal protective equipment (PPE failure). Table 1 illustrated the basic information about 55 participants’ performance in the recognition of 3 hazards. Most participants would choose to search the hazard in a sequential visual path, which presents a relatively higher average accuracy compared to those participants that response immediately capture only 1 or even less AOI. According to the sequence presence in the visual path, the participants were categorized as “sequence presence” group and “sequence absence” group in each hazard. For example, 90.91% of the participants made their judgments by observing multiple AOIs in Hazard #1, which implies that there exists the sequence between AOIs in their visual scanning path.

### 4.2. Hazard #1: Potential Electrical Contact

The first category of hazard was potential electrical contact, as shown in Figure 3, which was recorded in safety inspection standards database as “construction site power distribution system does not meet the requirements of three-level power distribution and two-level leakage protection”. The area of interest (AOI) “A”, denoted as AOI-A thereafter, is the distribution box; the AOI-B is the electrical wire connected to the box, and the AOI-C is the component connecting the box to the wall. The three AOIs A/B/C and the observation sequences between them were tested for necessity as condition variables, as shown in Table 2. Both A and B are necessary conditions for correct identification of the hazard, that is the core area in the visual scan path. No observation sequence between AOIs could be regarded as necessary sequence that can guiding to correct identification performance. Participants focused on the distribution box (A) or the electrical wires (B), and thus, successfully identified the hazardous “potential electrical contacts”. However, knowing only which of the AOIs A/B, or even C were observed provides limited clues about how the decisions of the presence of safety hazards were made. Further study of the visual patterns between them is worthy and necessary to be conducted.

Table 3 shows 7 visual search configurations of 55 participants’ hazard recognition process. For instance, Configuration #3 includes three visual search conditions: AC, BA, BC. According to Boolean operations, when all 3 sequential conditions were true simultaneously, a unique configuration, i.e., BAC, could be derived. A total of 6 participants adopted this visual configuration (N = 6), which reaches a PRI consistency of 83% (a configuration is considered effective with PRI consistency >75%). Configurations containing more than (not including) two participants will be considered non-contingent. Therefore, Configurations #1 #2, #3, #4 were included in the subsequent configuration analysis to summarize visual pattern solutions.

TQCA was performed to summarize the corresponding visual patterns, as shown in the Table 4. The results presented two main paths for potential electrical contact hazards, namely AB and BA. A total of 81.4% of these participants adopted these configurations and 85.3% of these participants correctly identified hazards. Particularly, 37.2% participants adopted configuration BA and 94.1% of these participants correctly identified hazards.

As shown in Figure 4, it was salient that workers tend to allocate priority attention to the intersection between energy-releasing source and wires. However, it is worth noting that according to the description in the safety inspection standards database, the distribution box seems to be an area containing more dangerous information. According to the results of post-experiment interview, participants believe that this area is the most common trigger area for safety hazards such as short circuit or electrical fire in engineering practice. Both AB and BA strategies can focus on this field in a limited time.

### 4.3. Hazard #2 and Hazard #3: Failure of Personal Protective Equipment

Numerous literature and statistics/reports indicate that a large part of unsafe behaviors that lead to accidents can be mainly summarized as failure of personal protective equipment (PPE) [2,30,33]. Although safety rules require workers to wear PPE on construction sites, such as helmets and safety harnesses when working at heights, studies have repeatedly shown that a significant number of injuries in the construction industry are due to workers not wearing their PPE.

#### 4.3.1. Hazard #2

The first PPE failure hazard was recorded in safety inspection standards database as “during the hanging basket construction of outdoor glass curtain wall, the workers did not wear safety helmet as required”, as shown in Figure 5.

In this hazard scenario, the AOI-A is the area above the worker’s chest, which contains the information that the worker is not wearing a helmet and is the core area of this hazard; corresponding to the AOI-A, the AOI-B is the area below the worker’s chest, which provides the information to determine the location and status of the worker; the AOI-C is the area of the outdoor glass curtain wall hanging basket operation, which implies a certain risk of falling objects to the workers below it. The results of condition necessity test are shown in Table 5. No necessary AOI and sequence condition was identified. Participants’ attention was relatively evenly dispersed among the three AOIs.

The visual patterns between these AOIs were further analyzed in the following configuration analysis. According to the Table 1, 78.18% of the participants chose to make judgment by observing multiple AOIs. The truth table (Table 6) shows all the visual sequences obtained from the visual tracking results. For example, Configuration 4 indicated that six participants (N = 6) adopted the “CA” visual sequence configuration with the consistency (reflecting the correct rate of hazard recognition) of 0.83. Similarly, the Configuration 7 of “ACB” (the unique sequence determined by the combination of the three sequence conditions that were all coded as “1”: AB, AC and CB) was adopted by four participants, corresponding to a 100% correct rate of hazard recognition.

TQCA analysis was performed, and the results of the intermediate solution are shown in the Table 7. Although the analysis results seem to present eight different visual path configurations, post-experimental interviews revealed that participants who correctly identified hidden hazards almost always did so based on the recognition of the non-helmeted hazard but differ in the most likely accident scenarios they envisioned, that are potential injuries from falling from platforms and from falling objects caused by the hanging basket or the objects it carries. Combining this information with a closer examination at the eight configurations, their key difference lies in the judgment of the AOI-A of this hazard. Accordingly, they could be grouped into two specific strategies.

The first strategy corresponds to the AB/ABC/ACB/AC configuration, in which participants will first target the area where the person is located, i.e., AOI-A, during the observation process, and focus on whether the construction workers wear personal protective equipment such as helmets in accordance with safety regulations, and then observe the surrounding environment and other components to complete a comprehensive judgment of the hazard. The second strategy corresponds to configuration CA/CBA/CAB/BC. Because of the existence of hanging baskets and high work situations in this scenario, participants’ feedback will first associate the potential risks of high work and falling objects, and then focus on workers who are not wearing helmets, thus completing the identification of hazards.

#### 4.3.2. Hazard #3

The second PPE failure hazard was recorded in safety inspection standards database as a “violation of safety operation regulations due to the absence of safety belts on workers performing work at heights”, as demonstrated in Figure 6.

In this hazard scenario, the AOI-A is the area where the workers working at height on the scaffold are located, contains the information that the workers are not wearing safety belts, which is the core components constituting this hazard; the AOI-B is the top area of the scaffold constituting the working environment at height, which is directly adjacent to the workers in space, and contains the clues for judging the location and status of the workers; the AOI-C is the middle support area of the scaffold, which helps determine the stability and structural soundness of the scaffold. The condition necessity test results were shown in Table 8.

The AOI-A is the only necessary basic AOI condition of the outcome, which contains the locations where worker in the construction site scene. This indicates that the area where worker is located is a necessary station in the visual scan path and the worker’s behavior or state is the primary area of attention allocation when identifying human-related FFH (fall from heights) hazards. The participants’ attention showed a very clear decreasing trend among the three AOIs, with PPE Failure in AOI-A, and the information of working at heights was contained in AOI-B receiving more attention. According to Table 1, 63.64% of the participants chose to make their judgments by observing multiple AOIs. Table 9 shows all the visual sequences obtained from the visual tracking results.

TQCA intermediate solution was shown in the Table 10. There are two main strategies: the first is the A–B sequence, which means a quick search of the human location area in the scene first, followed by a combination of environmental information to determine the existence and severity of hazards. The second strategy is B–A sequence, which means firstly forming a general impression of the overall scene, considering people as one of the elements in the scene in this stage, but not focusing on them first; then observing the state of people in the scene to make a comprehensive judgment. Corresponding to this hazard scene, that is, firstly identify and determine the working environment of scaffolding as a high working environment, and then combine with the objective fact that workers are in this environment without wearing safety belts, and make the judgment that hazard exists.

## 5. Discussions

### 5.1. Practical Contribution: Visual Cognitive Strategy for Hazard Recognition

The results shows that most participants chose the observation pattern of observing multiple AOIs in sequence before making judgments, presenting a high response accuracy rate.

#### 5.1.1. Potential Electrical Contact Hazards

As shown in Table 4, participants mainly adopted two strategies when observing potential electrical contact hazards: 32.6% of participants followed the observation pattern of “charged body (A)—wire (B)—connection between charged body and electric wire (hereinafter referred to as the intersection)”; 48.8% of participants conformed to the observation pattern of “wire (B)—energy-releasing source (A)—intersection”, which was followed by 48.8% of the participants.

It is worth noting that in the two different visual paths, the areas of charged objects (A) and wires (B) are both necessary conditions for participants to successfully recognize hazards. However, the results of heat maps and post-experiment interviews with participants showed that they were more concerned about the intersection of (A) and (B). On construction sites, the most common electrical fires come from the connection of energized wires to equipment [55,56], which is the intersection in the Hazard #2 scene. Interestingly, the interview results indicate that both groups of construction workers who adopted different observation sequences expressed confidence in this strategy and efficiency in choosing to focus on the intersection interface in this scenario. Although this common tendency provides specific guidance for interpreting construction workers’ recognition process of potential electrical contact hazards, it seems to be inconsistent with the priority areas of concern (e.g., charged bodies, ground wire) that have been generalized in previous studies [17,57]. The reason is probably that the participants in this study are all experienced construction workers, whereas these previous studies used alternative groups in the selection of participants, such as civil engineering students, who did not have as much experience as workers with work experience [17,58].

Therefore, the current research results show that when recognizing unknown scenes with potential electrical contact hazards, compared with energy-releasing source, giving priority to the intersection that considers the focus of attention among the components is more consistent with the cognitive patterns of safety-experienced construction workers, which supplements and revises the visual path generalized in previous visual research. Computer vision technology should pay more attention to the definition and feature extraction of the intersection areas to enhance the hazards recognition ability and efficiency in potential electrical contact hazard scenarios, as this may reflect the interaction between the two common components of energy-releasing source and wires. As RBC and gestalt model suggests that humans can identify hazards based on the spatial relationships of different elements and considering the extent to which they “deviate from the prototype” [18]. Further development and refinement of corresponding computer vision auxiliary technology aiming at hazard monitoring, such as the detection of the integrity of the insulation portion of the wire envelope, and the detection of abnormal heat generation, will occur in conjunction with thermal imaging technology [59].

#### 5.1.2. PPE-Related Hazard

Based on the results of data analysis, combined with post-experiment interviews, there are two strategies for participants to observe and recognize Hazards #2 and #3: (1) “Construction worker-PPE-construction work environment”. The corresponding specific visual search paths are shown in Solution #2-1 in Table 7 and Solution #3-1 in Table 10. (2) “Construction operating environment (association of potential hazards)—construction workers-PPE”. The corresponding specific visual search paths are shown in Solution #2-2 shown in Table 7 and Solution #3-2 shown in Table 10.

Safety training provides workers with the knowledge base and long-term memory of safety regulations that guide workers to the most likely interpretation and identification of a given scenario and generate perceptual goals. Construction workers are searching for hazards in accordance with this paradigm, which is the “top-down” model in psychology. The human-object-environment strategy described above is also consistent with the requirement in [OSHA] [60] that safety training prioritize whether workers working at heights are wearing safety devices. When the background (objects, features or groups in the scene) is consistent with the perceptual target, the effect of hazard recognition can be achieved [18]. In these two PPE-related hazard scenes of this study, the correct answer rate of construction workers using this paradigm in the two PPE hazard scenarios both exceeded 80%.

Even though the purpose of safety training is to reinforce long-term knowledge and enhance the effect of “top-down” inspection, due to the heavy construction workload, workers’ general perception starts from trivial signs and follows “heuristic” reasoning [61,62] to determine whether there are hazards, which is the “bottom-up” dominant cognitive model. This conforms to what psychology describes as the spontaneous interchange of “bistable perception” [63], which unconsciously shifts attention to prominent visual features of potential importance [64,65] that is, hazards closely related to workers [66]. This observation mode can be vividly described as “scene association type”. Although this bottom-up cognitive model may lay the foundation for understanding people’s selection of AOI, it is not sufficient to complete the visual searching cognitive process, especially in the dynamic construction site with high complexity. In such scenarios, potential hazards may come from many different and unintended areas. Therefore, the completion of the cognitive process also requires the supplement of associations with the current scene. Human perceptual mechanisms discard redundant information [67] and use existing information such as common sense, knowledge and experience, combined with their expectations of the current situation [68], to associate and discover missing information [18]. For example, under Hazard #2 and Hazard #3, when they quickly focus on the workers from the observation of the scene in hazard search tasks, they will generate expectations of the safety of the workers, that is, the workers should be protected by PPE. When further observations reveal that PPE is in a state of absence, they will make the judgment that workers are in a hazardous state, and the cognitive process of hazard recognition will be completed. This model can be summarized by “norm-guided”.

Interestingly, the results of the current study show that both strategies correspond to a fairly high correct rate of hazard recognition, without presenting a clear superiority or inferiority. However, it can be confidently concluded that the high accuracy rate corresponding to the normative interpretation means that the hazard recognition strategies taught in the safety training are effective, which verifies the positive effect of safety training on safety management. Safety training enables individuals to recall the requirements of safety regulations in construction scenarios, and consciously perform corresponding safety inspection and CHR. It is worth noting that although the “scene-associative” visual cognitive strategy appears to be less efficient than the “norm-guided” approach in targeting hidden areas, they both exhibit high levels of correctness, suggesting that the “scene-associative” may have potential implications for safety training. Specifically, the scenario-associative strategy is more reflective of worker experience, and although there may be significant internal individual variability, the overall consistent strategy presented helps to examine inappropriate instructional content that may be contrary to subjective human experience and habits in traditional construction safety training design [69]. Quality safety training content should include hands-on activities for visual exploration of the environment, or at least activities that provoke reflection on the possible sources of hazards, rather than merely conveying guiding information about visual exploration behavior that meets the normative guidance and requirements. Visual research can record reasonable and effective hazard recognition experience with common features through the cognitive strategies reflected in the visual sequence, which provides a rich source of content design for safety training to enhance its positive contribution to safety management [70,71].

In addition, these two strategies also have enlightening significance for the development of computer vision in hazard recognition. For example, the paradigm of human interpretation of safety specifications and the top-down visual cognitive strategy can be referred to learn how to accurately map the content in safety specifications with inherent complexity in specific scenarios [30], thus enhancing the computer’s ability to learn and understand safety regulations. The large amount of data recorded in vision research, after certain screening to optimize the quality of the data, can be used for the learning and training of computer vision algorithms [31], thus training computers to predict possible future hazardous states of the current scene, which may be significant for safety management. This is because what can be seen and recorded is the world composed of objects and surfaces, but humans can make perceptions and associations about the meaning of the observed visual attributes. The human cognitive system uses heuristic problem-solving shortcuts to make inferences about the received information for generating perceptions of the world rather than exhaustive algorithms [40,41]. In the current application paradigm of computer vision, searching for the presence of a feature is faster than for the absence of a feature [18], which is also the reason why a static scene with potential hazards may be more difficult to be recognized by computer vision than a dynamic scene. However, with the help of appropriate cues, things objectively absent in a scene can be interpreted as present but hidden, thus allowing for the simulation of human associative abilities. Vision research may provide such cues and learning basis for computer vision algorithms, such as the visual behavior corresponding to the visual cognitive strategy summarized in this study, which will bring leap-forward improvement for the hazard recognition ability of computer vision.

### 5.2. Theoretical Contribution

Constrained by the limitations of the analysis method, the construction industry study failed to accurately capture the temporal sequential visual patterns of CHR. This study introduces the TQCA method to summarize the visual sequence configurations with high identification correctness among all participants’ visual data, and summarize their strategies in combination with post-experiment interviews. The QCA method is highly applicable in revealing the influence of complex relationships among multiple antecedents on the results, given its ability to integrate the advantages of qualitative and quantitative analysis. On this basis, TQCA further extends the temporal dimension. By standardizing the encoding and analysis of visual sequences, TQCA substantially reduces the uniformity defects and substantial information loss caused by simple clustering [16] and neural network methods [17] used in previous studies, and also avoids the problems of excessive redundancy and complexity caused by quantitative methods such as recurrence quantification analysis [72] or multi-match (a vector-based multidimensional approach) [73,74,75].

This study extends and deciphers the mechanism of applying RBC and gestalt model to the construction hazard recognition. Construction industry research has long considered hazard as a combination of individual components (human, machine, object, etc.) [76], but the “intersection” identified in this study, as one of the interaction patterns between components, is an important cue emphasized by RBC and gestalt model. Similarly, the association of potential hazards formed by construction practitioners based on their observation of current scenarios and safety knowledge or experience is an important paradigm for the embodiment of gestalt model in the cognitive behavior of specific hazard recognition. Such an interpretation expands the specific paradigm of the application of the RBC and gestalt model in the process of CHR, providing concrete examples of the interaction between components and their possible role in the human cognitive process. Therefore, this implies that computers should consider more abstract spatial topological relationships such as interactions among components or clues to various potential hazard scenarios they imply, in addition to the traditional identification of people, machines, and objects when identifying hazards in the future. In addition, these findings expand the perspective of construction safety training bring potential for improving the reliability of safety training on construction site. 

## 6. Conclusions

This study aims to identify the temporal visual patterns exhibited by construction workers during their visual searching process and to explore the cognitive strategies of CHR reflected by these identified visual patterns through an eye-tracking experiment with real construction scene pictures as the background for CHR. The results show that: (1) In the potential electrical contact hazards, the intersection of the energy-releasing source and wire that reflected their interaction is the cognitively driven visual area that participants tend to prioritize; (2) in the PPE-related hazards, two different visual strategies, i.e., “scene-related” and “norm-guided”, can usually be generalized according to the participants’ visual cognitive logic in the recognition of PPE hazard, corresponding to the bottom-up (experience oriented) and top-down (safety knowledge oriented) cognitive models. This research furnishes a novel paradigm for the identification of visual patterns and the interpretation of CHR strategies from a cognitive perspective, as well as complements the RBC and gestalt model in visual cognitive theory, thus contributing to provide viable practical guidelines for the design and improvements of construction safety trainings and theoretical foundations of computer vision techniques for CHR.

There are still some limitations of this study that need to be considered. First, the hazard scenes were presented on the monitor in the form of 2D pictures, and although the material was derived from photos of the construction site, there was a gap between the perception and the real construction site. Second, the types of hazards involved in this experiment were limited to three. However, there are many types of hazards involved in construction projects, even including combinations of multiple hazard types. Therefore, the visual recognition strategies obtained in this study may not be applicable to other hazard types. Third, two parts of experiments, A and B, were designed in this study. However, the observation time of 3 s per picture in Part A was not sufficient to form a reliable visual strategy. Although this design utilized principles such as priming effects and working memory (short-term memory), it could be improved in subsequent studies. In the end, the practical implications of this study including the development of hazard recognition devices and employee safety training need further demonstration.

Based on the above limitations, this study suggests four potential valuable directions for future research. First, introduce virtual reality or augmented reality technologies into the research field of visual search sequence. Future studies may consider trying to use immersive and realistic reproduction of 3D construction scenes for the study if the technology allows. As a result, the data obtained from the study will be more realistic and reliable. Next, continue to expand the types of hazards involved in the experiments. Future studies are recommended to select more hazard scenarios and sample datasets to confirm the results and further explore the relationship between different hazard types as well as hazard recognition strategies. Then, in future studies, longer observation times for individual hazard scenes could be considered, thus, allowing participants to form strategies in a single observation. At last, researchers need to verify the practical implications of our findings through further experiments. For example, researchers are recommended to develop hazard recognition devices based on the visual cognitive strategies and select sample construction project to explore the practical value of this study for employees’ safety training design.

## Figures and Tables

**Figure 1 ijerph-18-08779-f001:**
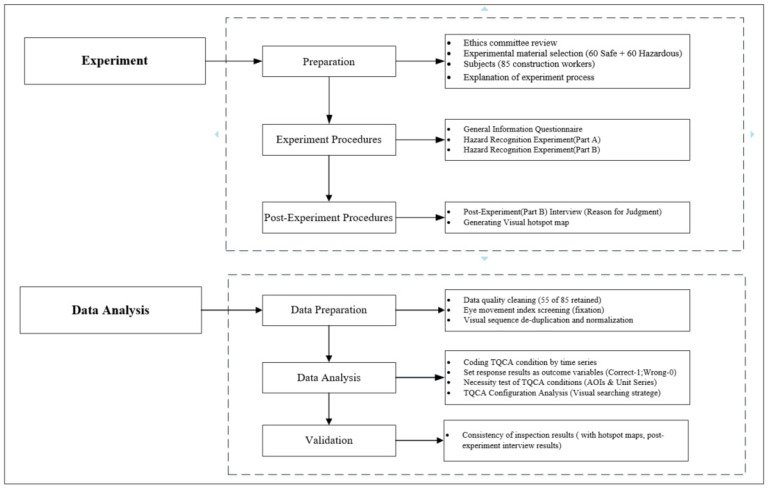
Graphic representation of the methodology.

**Figure 2 ijerph-18-08779-f002:**
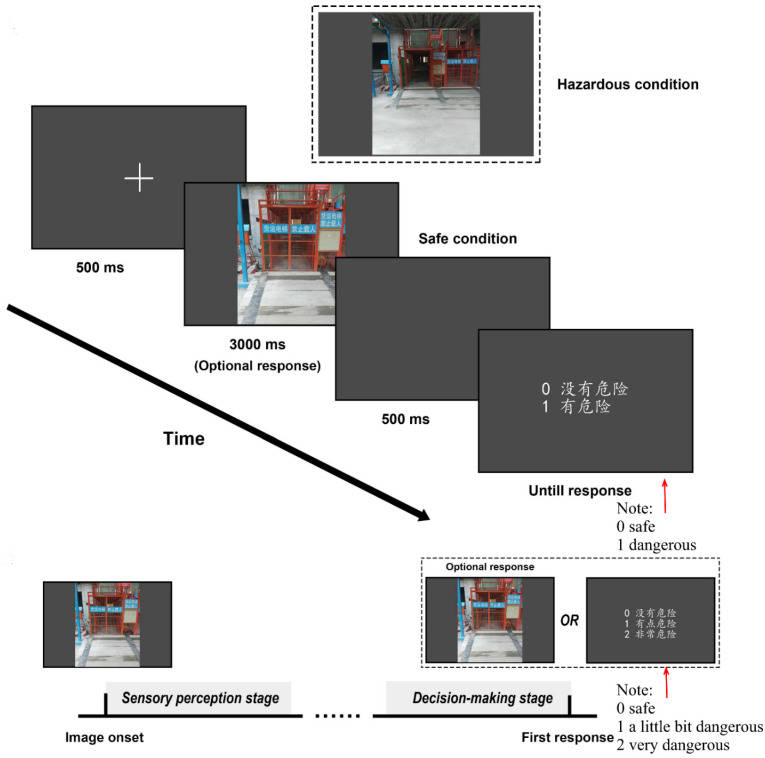
Experimental design. Note: The direction of the arrow represents the experimental flow. For specific experimental procedures, please refer to the first paragraph of Section 3.2.1.

**Figure 3 ijerph-18-08779-f003:**
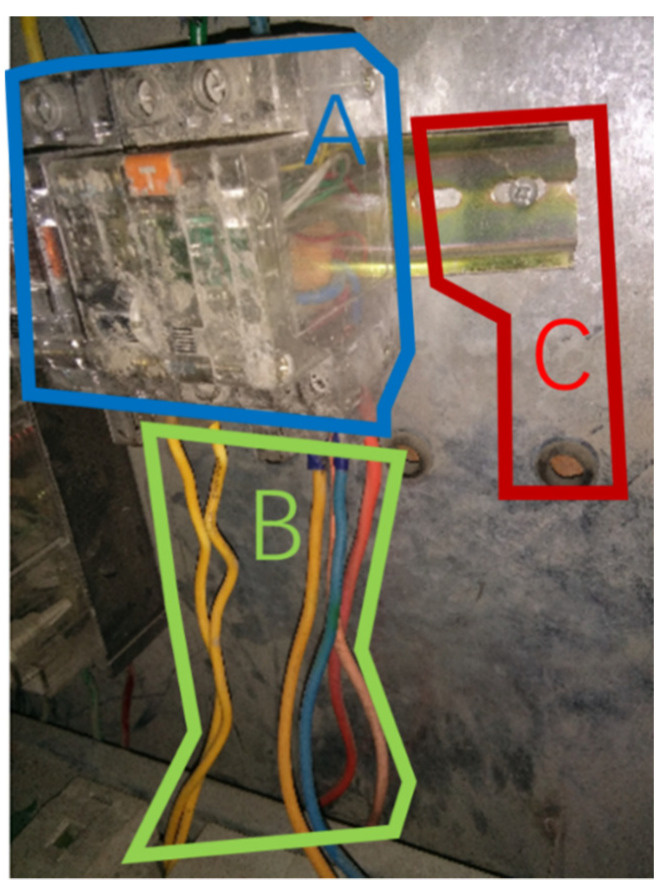
Potential electrical contact (Hazard #1).

**Figure 4 ijerph-18-08779-f004:**
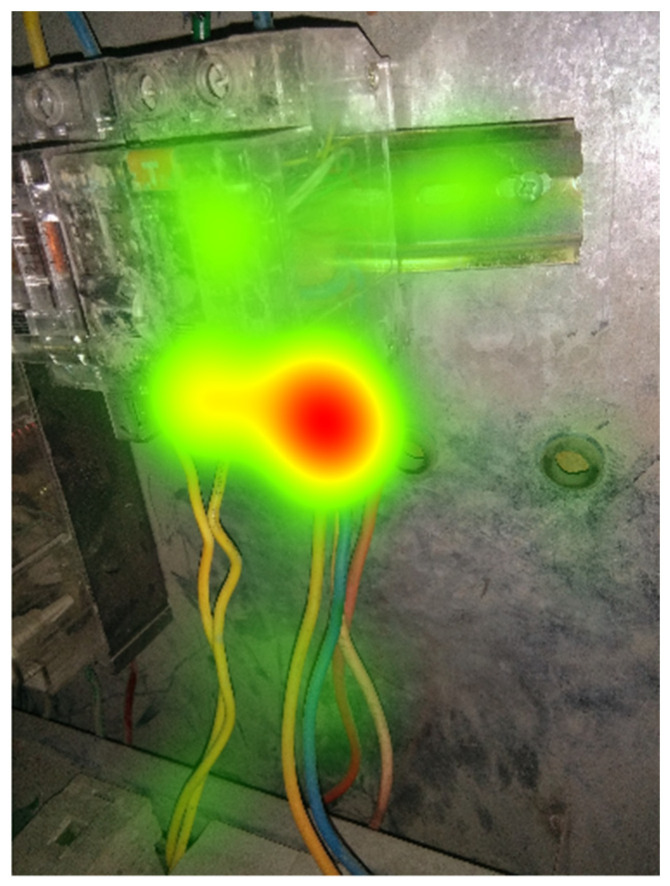
The visual heat map of Hazard #1.

**Figure 5 ijerph-18-08779-f005:**
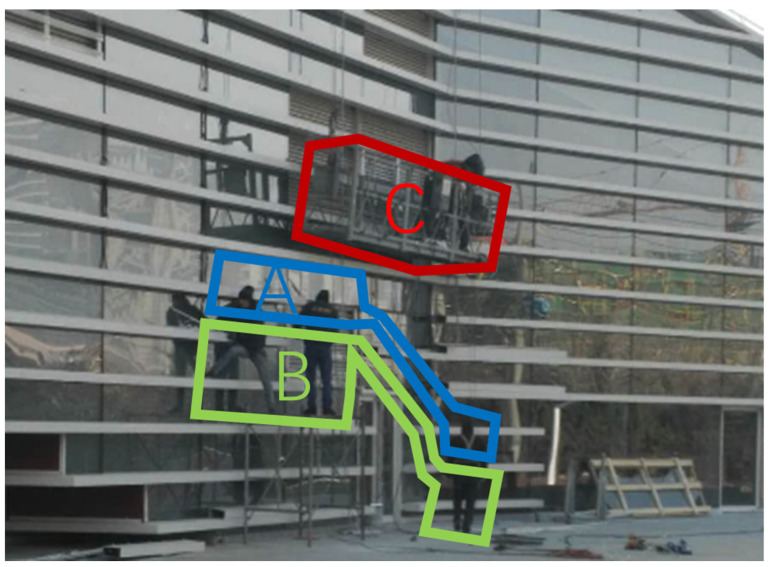
PPE failure (Hazard #2).

**Figure 6 ijerph-18-08779-f006:**
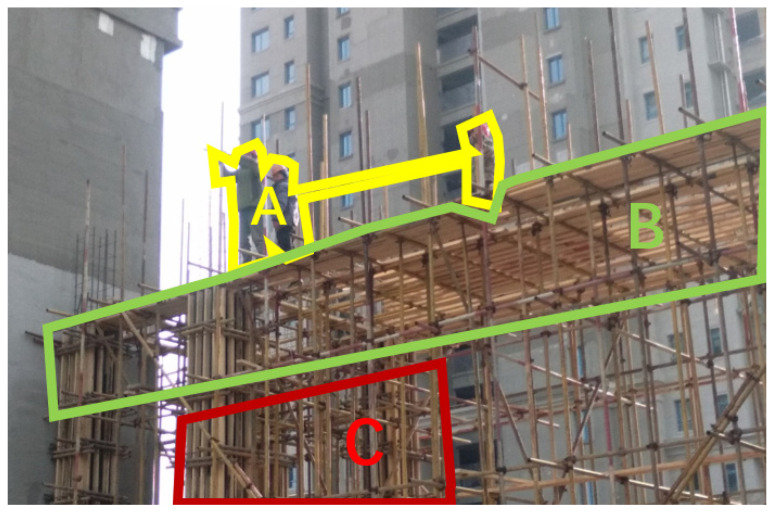
PPE failure (Hazard #3).

**Table 1 ijerph-18-08779-t001:** Statistics of participants’ performance in the recognition of three hazards.

Hazard Type	Hazard	Group	Number	Percentage	Accuracy
Potential Electrical Contact	Hazard #1	Sequence Presence	50	90.91%	80.00%
		Sequence Absence	5	9.09%	60.00%
		ALL	55	100.00%	78.18%
PPE Failure	Hazard #2	Sequence Presence	43	78.18%	95.35%
		Sequence Absence	12	21.82%	66.67%
		ALL	55	100.00%	89.09%
	Hazard #3	Sequence Presence	35	63.64%	91.43%
		Sequence Absence	20	36.36%	90.00%
		ALL	55	100.00%	90.91%

**Table 2 ijerph-18-08779-t002:** Results of necessity tests (Hazard #1).

Outcome Variable: 1		
Conditions Tested	Consistency	Coverage
A	0.953	0.788
B	0.953	0.804
C	0.256	0.687
AB	0.395	0.739
AC	0.186	0.727
BA	0.535	0.852
BC	0.209	0.642
CA	0.047	0.500
CB	0.023	1.000

**Table 3 ijerph-18-08779-t003:** The visual scan path configuration (Hazard #1).

Configuration		Sequence Conditions						Outcome	N	PRI Consistency	Effective?
		AB	AC	BA	BC	CA	CB				
#1	(CBA)	0	0	1	0	1	1	1	1	1.00	Y
#2	(BA)	0	0	1	0	0	0	1	17	0.94	Y
#3	(BAC)	0	1	1	1	0	0	1	6	0.83	Y
#4	(AB)	1	0	0	0	0	0	1	18	0.78	Y
#5	(-)	0	0	0	0	0	0	0	5	0.60	N
#6	(ABC)	1	1	0	1	0	0	0	5	0.60	N
#7	(BCA)	0	0	1	1	1	0	0	3	0.33	N

**Table 4 ijerph-18-08779-t004:** Intermediate solution to correct recognition of Hazard #1.

Intermediate Solution Frequency Cutoff: 3; Consistency Cutoff: 0.75					
Solutions		Configurations	Raw Coverage	Unique Coverage	Consistency
#1	AB	AB	0.326	0.326	0.78
#2	BA	BA	0.372	0.372	0.941
		BA&BC&AC	0.116	0.116	0.833
Overall solution coverage: 0.814					
Overall solution consistency: 0.853					

Note: Raw coverage: total share of the outcome that is explained by a configuration. (The total proportion of participants adopting the solution). Unique coverage: unique share of the outcome that is explained by the configuration. (The unique proportion of participants adopting the solution). Overall solution coverage: the extent to which the cases within the dataset correspond to the relationships in the solution [53]. (The sum of the proportion of participants corresponding to all solutions). Overall solution consistency: a measure of fitness for the entire set of configurations, which makes it analogous to R-squared in regression [54]. (The overall correct rate of all participants corresponding to all patterns in the table).

**Table 5 ijerph-18-08779-t005:** Results of Necessity Tests (Hazard #2).

Outcome Variable: 1		
Conditions Tested	Consistency	Coverage
A	0.755	0.925
B	0.653	0.970
C	0.816	0.976
AB	0.408	1.000
AC	0.327	1.000
BA	0.143	0.875
BC	0.265	1.000
CA	0.327	0.941
CB	0.306	1.000

**Table 6 ijerph-18-08779-t006:** The visual scan path configuration (Hazard #2).

Configuration		Temporal Sequence Condition						Outcome	N	PRI Consistency	Effective?
		AB	AC	BA	BC	CA	CB				
#1	(-)	0	0	0	0	0	0	0	12	0.67	N
#2	(ABC)	1	1	0	1	0	0	1	7	1.00	Y
#3	(CAB)	1	0	0	0	1	1	1	6	1.00	Y
#4	(CA)	0	0	0	0	1	0	1	6	0.83	Y
#5	(AC)	0	1	0	0	0	0	1	4	1.00	Y
#6	(BC)	0	0	0	1	0	0	1	4	1.00	Y
#7	(ACB)	1	1	0	0	0	1	1	4	1.00	Y
#8	(CBA)	0	0	1	0	1	1	1	4	1.00	Y
#9	(AB)	1	0	0	0	0	0	1	3	1.00	Y
#10	(BA)	0	0	1	0	0	0	0	2	0.50	N

**Table 7 ijerph-18-08779-t007:** The visual scan path configuration (Hazard #2).

Intermediate Solution Frequency Cutoff: 2; Consistency Cutoff: 0.75				
Solutions	Configurations	Raw Coverage	Unique Coverage	Consistency
#1	AB	0.061	0.061	1.000
	ABC	0.143	0.143	1.000
	ACB	0.082	0.082	1.000
	AC	0.082	0.082	1.000
#2	CA	0.102	0.102	0.833
	CBA	0.082	0.082	1.000
	CAB	0.122	0.122	1.000
	BC	0.082	0.082	1.000
Overall solution coverage: 0.755.				
Overall solution consistency: 0.974				

**Table 8 ijerph-18-08779-t008:** Results of necessity tests (Hazard #3).

Outcome Variable: 1		
Conditions Tested	Consistency	Coverage
A	0.940	0.922
B	0.660	0.917
C	0.040	1.000
AB	0.200	0.833
AC	0.040	1.000
BA	0.440	0.957
BC	0.040	1.000
CA	0.000	0.000
CB	0.000	0.000

**Table 9 ijerph-18-08779-t009:** The visual scan path configuration (Hazard #3).

		Temporal Sequence Conditions									
Configuration		AB	AC	BA	BC	CA	CB	Outcome	N	PRI Consistency	Effective?
#1	(BA)	0	0	1	0	0	0	1	0.95	0.95	Y
#2	(-)	0	0	0	0	0	0	1	0.90	0.90	Y
#3	(AB)	1	0	0	0	0	0	1	0.83	0.83	Y
#4	(BAC)	0	1	1	1	0	0	1	1.00	1.00	Y

**Table 10 ijerph-18-08779-t010:** Intermediate solution to correct recognition of Hazard #3.

Intermediate Solution Frequency Cutoff: 12; Consistency Cutoff: 0.75				
Solutions	Configurations	Raw Coverage	Unique Coverage	Consistency
#1	AB	0.200	0.200	0.833
#2	BA	0.400	0.400	0.952
Overall solution coverage: 0.600				
Overall solution consistency: 0.909				

## Data Availability

Our data are available upon request from the corresponding author.

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
