# Peer review of "Temporal Visual Patterns of Construction Hazard Recognition Strategies"

_ijerph, 2021, doi:10.3390/ijerph18168779_

Round 1

Reviewer 1 Report

This manuscript, titled “Re-investigation of Normative Visual Patterns of Occupational Hazard Recognition,” analyzed the construction hazard recognition visual searching sequences using the temporal qualitative comparative analysis (TQCA) approach. The paper is well-written, the structure is well-organized, and the content has a good logical flow.

I have several suggestions to further improve the paper.

First, Section 2.2 can be enriched by comparing TQCA approach with the three approaches discussed. Section 1 can be trimmed down to make space for such an addition.

Second, some clarity is needed for lines 223-227. It is still not clear what is the issue with Part B. Also, based on the vague understanding of lines 223-227 that I managed to gather, the working memory explanation does not seem to be applicable in this case. Perhaps priming effect or short-term memory explanation works better.   

Third, a discussion on the limitation and suggestions for future studies would add more value to the paper.

Author Response

Dear reviewer, please see the attachment for our reply to your valuable comments.

Reviewer 2 Report

First of all, I would like to thank you for the opportunity to consider the article "Re-investigation of Normative Visual Patterns of Occupational Hazard Recognition". The ability of a person to respond to an emerging hazard situation is one of the important aspects in assessing risks and selecting appropriate preventive measures. The visual searching process in construction hazard recognition, recorded by an eye-tracking experiment, and the TQCA method is undoubtedly an interesting approach to modeling and verifying a person's ability to prevent accidents.

I have the following comments on the article:

  1. In the abstract, lines 17, 18, you describe the analyzed hazards and results with numbering 1) and 2). It should not be in the abstract, the solved problem should be briefly described.
  2. Has the hazard recognition experiment been focused only on the assessment of the source of danger or does it also make it possible to analyze the recognition of the occurrence of a hazard situation in a given activity (so-called accident scenario)?
  3. Line 215, Figure 2. It is not clear from the figure what the values ​​of 0.1, 0, 1, 2 mean as they are not in English. It would be appropriate to add a note - explanation to the title of the image.
  4. Line 286 typo at the end of the sentence (.,).
  5. Line 278, 305, 378 are missing a space between the word and the citation in parentheses. Similarly, Line 502 between the word "source (A)".
  6. Line 282, I recommend dividing the sentence "... use of QCA methods. QCA has often ... ".
  7. Line 391 you refer to “... numerous literature and statistics/reports...” but you have no reference to any literature there. It would be appropriate to add at least some resources.
  8. Line 428, you are talking about "hidden hazards". Are you sure that they are "hidden“ when you monitored the ability to recognize hazards with human own senses in your research?
  9. Line 466, what are FFH hazards?
  10. Typo: Dot at the end of a sentence, after a reference to the literature.

Author Response

(The authors gave the same response as above.)

Reviewer 3 Report

With this work, a visual searching process in construction hazard recognition was recorded by an eye-tracking experiment, and a Temporal Qualitative Comparative Analysis (TQCA) method was applied to conduct detailed necessity tests and configuration analysis of visual sequences to summarize and interpret the visual searching strategy of the construction hazard recognition process for the development of Construction Hazard Recognition (CHR) computer vision techniques and the design of safety training.

This is an interesting paper on which I have some comments. I recommend that authors consider my comments to improve their manuscript mainly in the introduction section and methodology section:

  1. Introduction section: Authors should clearly define the research objective and problem (it seems that the objective is described only in the Discussion section). In this regard, there is much dispersion throughout the manuscript. Regarding the research problem, the authors point out methodological deficiencies and limitations in CHR visual patterns studies. However, the authors do not specify what problems they address in this work.
  1. Methodology section: in my opinion, the experiment has not been explained correctly. Essentially, the authors have designed the eye-tracking experiment based on different cognitive models and paradigms (long-term memory, working memory, and perception) as well as the scanpaths in eye movements during such perception. However, this approach and integration of models and paradigms has not been well explained and argued. For example: (i) the experiment regarding parts A and B is not comprehensible. Please explain the experiment in the necessary detail (Figure 2 is not cited in the text. It is probably an error in line 197); (ii) it is advisable to explain in more detail the implication of working memory and, where appropriate, the priming effect. To do so, consider the time variable and the necessary bibliographic citation; (iii) the description and application of the statistical analysis should be substantially improved. Please, the sequence exposed in 3.4.2 section and 4. section should be clear.

In summary, the authors must mainly improve: (i) the objective and justification of the research; (ii) the methodology used, considering for this the integration of models and paradigms typical of cognitive psychology and occupational safety; (iii) statistical analysis. All this could modify the discussion section and conclusions section.

Author Response

(The authors gave the same response as above.)

Round 2

Reviewer 3 Report

First of all, I would like to acknowledge the effort made by the authors in the development of their research, as well as in the revision of the manuscript.

In relation to the current version of the manuscript, in my opinion, it is not supported by a sufficiently solid structure. I continue to observe this weakness, mainly through the common thread that relates the title, objectives, methodology, and conclusions.

The title is not consistent with the objectives and neither with the conclusions. The objectives continue to be poorly written. In this regard, among other aspects, there are conceptual differences between what is indicated in lines 542-543 and in the introduction section.

In any case, the objectives pursued by the authors are not clear. These objectives should have a clear connection and alignment with section 2.2 on Methodological Deficiencies and Limitations as well as with the discussion and conclusions.

The conclusions add further confusion to the above, as it does not respond to the objectives and questions posed in the previously cited sections.

Author Response

Dear reviewer, please see the attachment for our reply. Thank you very much!
